# Purification and Characterization of a Novel Alginate Lyase from a Marine *Streptomyces* Species Isolated from Seaweed

**DOI:** 10.3390/md19110590

**Published:** 2021-10-20

**Authors:** Thi Nhu Thuong Nguyen, Timothy Chataway, Ricardo Araujo, Munish Puri, Christopher Milton Mathew Franco

**Affiliations:** 1Department of Medical Biotechnology, College of Medicine and Public Health, Flinders University, Adelaide, SA 5042, Australia; nhuthuongnt@ntu.edu.vn (T.N.T.N.); ricjparaujo@yahoo.com (R.A.); munish.puri@flinders.edu.au (M.P.); 2Department of Biotechnology, Institute of Biotechnology and Environment, Nha Trang University, Nha Trang 650000, Khanh Hoa, Vietnam; 3Proteomics Facility, College of Medicine and Public Health, Flinders University, Adelaide, SA 5042, Australia; tim.chataway@flinders.edu.au; 4i3S-Instituto de Investigação e Inovação em Saúde, University of Porto, 4200-135 Porto, Portugal; 5INEB-Instituto Nacional de Engenharia Biomédica, Universidade do Porto, Rua Alfredo Allen, 208, 4200-180 Porto, Portugal

**Keywords:** actinobacteria, alginate lyase, polysaccharide-degrading enzyme, protein sequence, bifunctional enzyme, seaweed

## Abstract

Alginate, a natural polysaccharide derived from brown seaweed, is finding multiple applications in biomedicine via its transformation through chemical, physical, and, increasingly, enzymatic processes. In this study a novel alginate lyase, AlyDS44, was purified and characterized from a marine actinobacterium, *Streptomyces luridiscabiei*, which was isolated from decomposing seaweed. The purified enzyme had a specific activity of 108.6 U/mg, with a molecular weight of 28.6 kDa, and was composed of 260 amino acid residues. AlyDS44 is a bifunctional alginate lyase, active on both polyguluronate and polymannuronate, though it preferentially degrades polyguluronate. The optimal pH of this enzyme is 8.5 and the optimal temperature is 45 °C. It is a salt-tolerant alginate lyase with an optimal activity at 0.6 M NaCl. Metal ions Mn^2+^, Co^2+^, and Fe^2+^ increased the alginate degrading activity, but it was inhibited in the presence of Zn^2+^ and Cu^2+^. The highly conserved regions of its amino acid sequences indicated that AlyDS44 belongs to the polysaccharide lyase family 7. The main breakdown products of the enzyme on alginate were disaccharides, trisaccharides, and tetrasaccharides, which demonstrated that this enzyme acted as an endo-type alginate lyase. AlyDS44 is a novel enzyme, with the potential for efficient production of alginate oligosaccharides with low degrees of polymerization.

## 1. Introduction

Alginate is the most abundant polysaccharide located in the matrix and in the cell wall of brown seaweed. It is composed of α-L-guluronate (G) and β-D-mannuronate (M) as the major monomeric units [1]. These units are organized in three different blocks: poly α-L -guluronate (polyG), poly β-D- mannuronate (polyM), and the heteropolymer (polyMG) [2]. The number of G and M blocks, as well as the ratio of G to M within alginates, are varied and depend on the sources from which they are isolated. Lee and Mooney [3] summarized the more than 200 different alginates that are currently being manufactured. The structure of alginate is variable, presenting two types of homopolymeric sequences (MM and GG), as well as heteropolymeric sequences (MG and GM) [4].

Alginates generally have molecular weights ranging from 500 kDa to 1000 kDa [5]. Alginates liberated from brown seaweeds comprise up to 40–47% dry biomass [6]. For example, the alginate contents are 22 to 30% of the dry weight in *Ascophyllum nodosum* and 25 to 44% for *Laminaria digitata* [2]. Alginate can be hydrolyzed to oligosaccharide alginates, which are gaining more and more attention due to their wide applications. They can be used as food, plant growth stimulants, and for various biomedical applications [7]. The biomedical applications are possible due to its biocompatibility and, in addition to tissue regeneration, include in situ drug delivery and wound healing, bone regeneration and cartilage repair, and drug delivery [8].

Characterized as being either mannuronate or guluronate lyases, alginate lyase cleaves the glycosyl linkages of alginate by the β-elimination mechanism, which results in the production of oligosaccharides with an unsaturated uronic acid [9]. Alginate lyases have been extracted from multiple sources, such as marine algae, marine molluscs (*Littorina* spp., *Turbo cornutus*, *Haliotis* spp.) [10], and a wide range of marine microorganisms (bacteria and fungi) and marine invertebrates. The following bacteria have been reported to produce enzymes that can degrade brown algae polysaccharides: *Alginovibrio aquatilis, Azotobacter vinelandii, Pseudomonas aeruginosa, Pseudomonas maltophilia, Flavourbacterium* sp., and *Vibrio* sp. [11]. Schaumann and Weide [12] demonstrated that there are four species of marine fungi which have the ability to degrade alginate and produce alginate lyase: the deuteromycetes *Asteromyces cruciatus, Dendryphiella arenaria,* and *Dendryphiella salina,* and the ascomycete *Corollospora intermedia*. In recent years, alginate lyase isolated from strains of marine microorganisms have been developed to produce novel polymers of alginate for various applications in the agricultural, industrial, and medical fields [13].

Alginate lyases can be classified into two groups according to their substrate specificities: there are polyG-block specific lyases and polyM-block specific lyases [10]. The majority of alginate lyases are observed to be polyM-specific lyases [14]. However, some alginates lyases can degrade both polyM and polyG-block (polyMG-specific lyase) and show a more bifunctional activity. Otherwise, alginate lyases can also be classified as endolytic and exolytic alginate lyases, based on their mode of action [1]. The glycosidic bonds broken down by endolytic alginate lyases within alginate polymers result in the liberation of the main products of unsaturated oligosaccharides (di-, tri-, and tetra-saccharides). Alternatively, exolytic alginate lyase can additionally degrade oligosaccharides, resulting in monomers; although in recent years, exo-type lyases have rarely been reported and characterized [15,16,17]. Therefore, finding new alginate lyases and their characterization would make possible the tailoring of alginate oligosaccharides to obtain targeted functional properties.

Actinobacteria are one of the major phyla of bacteria that have been isolated from a wide range of marine samples. One of the genera of actinobacteria that has the largest number of species identified to date is *Streptomyces*, which possess many important features. In addition to secondary metabolites, including antibiotics, *Streptomyces* are producers of industrially important enzymes, such as amylase, cellulase, gelatinase, casein hydrolysate, chitinase, and lipase [18,19,20,21,22]. These enzymes can be widely applied in biotechnology and, specifically, in the nutrition and biomedical fields [23]. In recent years, marine actinobacteria have been reported to produce enzymes that degrade polysaccharides such as alginate, laminarin, and fucoidan [13]. There are many avenues for the exploitation of the treated alginate and their degradation products [24,25,26].

In this work, a novel alginate lyase from a marine actinobacterium was identified and purified. Molecular identification of the bacterial strain for the production of alginate lyase was investigated. Its properties, in terms of enzyme activity and amino acid sequence, were analyzed. The effects of various factors, including temperature, pH, and metal ions, on enzyme activity were also investigated. This study is the first report on a bifunctional alginate lyase from *Streptomyces luridiscabiei*. The production of bacterial alginate lyase from a wild strain and its identification is a key step in establishing its biotechnological application for seaweed processing and for controlled alginate hydrolysis.

## 2. Results and Discussions

### 2.1. Identification of Marine Actinobacterium

Eighty strains of marine actinobacteria isolated from decomposing seaweed, including brown, green, and red seaweed, were screened for their ability to grow on bull kelp powder (2%, *w*/*v*), as the sole carbon source present in the growth medium. Only 12 isolates showed growth on medium containing seaweed after 7 days of incubation at 27 °C, with shaking at 250 rpm. These strains were evaluated for their ability to hydrolyze alginate and strain DS44 was selected because it exhibited the strongest alginate lyase activity.

The results of the whole genome sequencing and 16S rRNA gene (Accession no. OK148322) showed that strain DS44 was identified as being closest to *Streptomyces luridiscabiei,* with a sequence similarity of 99.9%, genome size of 13.28 MB, and G + C content of 67.6%. Therefore, this strain is a member of the genus *Streptomyces*.

### 2.2. Purification of AlyDS44

The crude extract was concentrated 25-fold by 60% ammonium sulfate precipitation, followed by dialysis, with a specific activity of 27.54 U/mg. The protein was further purified by FPLC on a Mono-Q column (GE Healthcare/Cytiva, Marlborough, MA, USA) and yielded a purification factor of 92.85, with a specific activity of 93.78 U/mg, while the percentage recovery was 4.15% (Table 1). The fractions containing the peak were pooled and further purification of those fractions was achieved by size exclusion chromatography (Superdex 75). The purified enzyme had a specific activity of 108.6 U/mg. These steps resulted in the purification of the alginate lyase, and each step of the purification was analyzed by SDS PAGE (Figure 1). It can be seen that the specific activity after purifying by anion exchange chromatography and size exclusion chromatography increased 94.4-fold over that of the crude extract. Purification of AlyDS44 resulted in a yield of 1.79% and a Specific Activity of 109 U per milligram.

The fractions purified by size exclusion chromatography were pooled, concentrated using an ultrafiltration membrane (Amicon Ultra-15, 10 kDa, Merck Millipore, Burlington, MA, USA), and stored at −20 °C for further use.

### 2.3. Characterisation of Enzymes

#### 2.3.1. Molecular Weight of Enzymes

The pure alginate lyase, AlyDS44, has an approximate molecular mass of 29–32 kDa and belongs to the low molecular weight group of 25–30 kDa [10], which is within the size range for alginate lyases produced by marine bacteria from 24 kDa to 110 kDa [11]. Alginate lyases with similar molecular weights were produced by *Microbulbifer* sp. ALW1 (26.2 kDa) [27], *Isoptericola halotolerans* CGMCC 5336 (28 kDa) [28], and *Streptomyces*. sp AGL-5 (27.5 kDa) [29]. Alginate lyases with higher molecular weights include AlyM from *Microbulbifer* sp.Q7 (63 kDa) [30], AlyA5 from *Zobellia galactanivorans* (69.5 kDa) [31], and AlgH1 from *Marinimicrobium* sp. H1 (61.3 kDa) [32].

#### 2.3.2. Determination of Amino Acid Sequence of AlyDS44

The purified AlyDS44 band was subjected to nanospray qTOF mass spectrometry (Sciex, Framingham, MA, USA) analysis, and 10 peptides were matched against the *Streptomyces sp. AVP053U2* database, with 99% confidence (Proteinpilot v4.5). The peptide sequences of length 12–26 amino were converted to DNA sequence by using a codon usage table for the G + C rich *Streptomyces* [33]. The alginate lyase gene was found by comparing the longest DNA sequence with the whole genome of *Streptomyces luridiscabiei* using Geneious software. After obtaining the full alginate lyase gene sequences, the complete amino acid sequences were found by using the translation tool ExPASy (Expert Protein Analysis System, Swiss Institute of Bioinformatics—www.epasy.org, accessed on 6 October 2021) to convert DNA to protein. The result showed that the AlyDS44 alginate lyase gene was composed of 780 bp (GenBank Accession No: OK169607; NCBI, Bethesda, MD, USA), encoding 260 amino acid residues. Each of the 10 peptides sequenced had 100% identity with the genetic sequence.

Alginate lyases can be classified into 14 Polysaccharide Lyase (PL) families, namely the PL5, PL6, PL7, PL14, PL15, PL17, PL18, PL31, PL32, PL34, PL36, and PL 39 families, based on the similarity of protein sequences and structural features [10,25,26]. The PL7 family contains the largest number of members of the enzyme, which are mostly produced by bacteria [34]. PL7 alginate lyase consists of three highly conserved regions, including R(S/N)ERL(E/A/V), Q(I/V)H, and YFKAG(A/G/N/V/L)Y [35]. Some alginate lyases of the PL7 family from *Vibrio* spp. and *Streptomyces* sp. ALG5, which have different substrate specificity, were compared with the identified amino acid sequences of the alginate lyase from strain DS44 to reveal the regions of similar sequences. The multiple sequence alignment of the catalytic domain of AlyDS44 (Figure 2) indicated that it contained three conserved regions of R(S/A)ERL, QIH, and YFKAG(A/G)Y. It was speculated that AlyDS44 belongs to the PL7 family. For the relationship between amino acid sequence and substrate specific, Zhu and Yin (2015) found that the polyM specific alginate lyase contained QVH, while the polyG specific alginate lyase contained QIH, in the conserved regions [10]. For example, alginate lyase A9mT of *Vibrio* sp. JAM-A9m and AlyVOA of *Vibrio* sp. O2 preferred to degrade polyM, containing QVH regions [36,37]. Otherwise, AlyVI from *Vibrio* sp. QY101 and AlgNJU-03 from *Vibrio* sp. NJU-03, possessing QIH in the conserved regions, were shown to have a higher specific activity for the polyG substrate [38,39]. The results of multiple sequence alignments of the identified amino acid sequences of AlyDS44 showed that the enzymes having a QIH motif are associated with a preference to degrade polyG. The structural relationship between AlyDS44 and the other characterized alginate lyases that belong to the PL6, PL7, PL15, PL17, and PL18 families was analyzed and is shown in the phylogenetic tree (Figure 3). It should be pointed out that the amino acid sequence of AlyDS44 showed more than 10% difference (Figure 2) compared with the reported most-similar sequence of the alginate lyase from *Streptomyces* sp. ALG-5 [29], which demonstrated that AlyDS44 alginate lyase from *Streptomyces luridiscabiei* is a novel enzyme. The enzyme properties of AlyDS44, including optimal pH, temperature, effect of NaCl, and elements, could not be compared with ALG-5 alginate lyase, as it has not been characterized.

### 2.4. Analysis of Degradation Products of DS44 Alginate Lyases by ESI-MS

The negative-ion electrospray ionization mass spectra of hydrolysis products of alginate by AlyDS44 is shown in Figure 4. The ions at m/z 373, 571, 769, and 967 exhibited unsaturated disaccharide, unsaturated trisaccharide, unsaturated tetrasaccharide, and unsaturated pentasaccharide, respectively [30].

After degrading for 24 h at 37 °C, the main AlyDS44 degradation products were determined as disaccharides, trisaccharides, and tetrasaccharides. The results verified that AlyDS44 alginate lyases cleaved the glycosidic bonds in alginate by β-elimination reaction and also indicated that this enzyme is an endo-type alginate lyase [30,32].

### 2.5. Optimisation of pH, Temperature, and NaCl, and the Effect of Metal Ions on Alginate Lyase Activity

The maximum activity of AlyDS44 was obtained at pH 8.5 in 0.02 M Tris-HCl buffer (Figure 5a). Greater than 70% of its maximum activity was exhibited in the pH range of 6.5 to 9.5. Some previous studies have shown that the majority of alginate lyase from marine bacteria have optimal activity between pH 7.0 and 8.5 [26,28]. The temperature at which the maximum activity of AlyDS44 was observed was 45 °C (Figure 5b). AlyDS44 alginate lyase presented over 80% relative activity in the temperature range of 35 °C to 55 °C.

There are significant differences in terms of the optimal temperatures of alginate lyases [26]. The optimal temperature of ALyDS44 was similar to that of *Microbulbifer* sp. ALW1, *Marinimicrobium* sp. H1, and *Shewannella* sp. YHI [27,32,40], but lower than that of *Sphingomonas* sp. MJ-3 and *Microbulbifer* sp. 6532A, which have an optimal temperature of 50 °C [40,41].

AlyDS44 alginate lyase was active in the presence of a wide concentration range of NaCl (0.2–1 M) and showed a highest activity of 125% in 0.6 M NaCl. Interestingly, AlyDS44 could maintain a high activity at a concentration of 1 M NaCl, which was similar to AlyAL-28 lyase extracted from *Vibrio harveyi* AL-28 [42]. Therefore, AlyDS44 also belongs to a group of salt-tolerant alginate lyases isolated from marine bacteria [27,32,43]. Therefore, AlyDS44 has the potential to be employed in a high-salt environment.

As shown in Figure 6, Mn^2+^, Co^2+^, and Fe^2+^ could enhance the alginate degradation activity of AlyDS44. These results were similar to reports of M3 lyase and alginate lyase from *Streptomyces* spp. [44,45] and the alginate lyase from *Isoptericola halotolerans* CGMCC 5336 [28]. Surprisingly, the activity of AlyDS44 more than doubled in the presence of Mn^2+^ and Co^2+^, with increases of 242% and 219% relative activity, respectively. Ca^2+^ and Mg^2+^ had no effect on the enzyme activities [27]. In contrast, AlyDS44 was clearly inhibited by Zn^2+^ and Cu^2+^, while Fe^3+^ showed a slight inhibitory effect, as observed for other alginate lyases [27,38,46]. Many of these metal ions are present in the media, either as trace elements or mineral ions, and thus should be used at a suitable concentration.

### 2.6. Hydrolysis of Alginate, and PolyM and PolyG Alginates

AlyDS44 exhibited the highest activity toward sodium alginate (Figure 7). Furthermore, purified alginate lyase could hydrolyze both PolyM and PolyG, indicating that they are bifunctional alginate lyases; though it degraded PolyG more efficiently than PolyM. The results indicated that AlyDS44 should be classified as a PolyG-block specific lyase. This result is consistent with the result of the multiple sequence alignments described in Figure 2. A large proportion of alginate lyases are PolyM-specific lyases [14,25,26]; thus, the enzymes which degrade PolyG more effectively are highly desirable for the modification of alginate. This characteristic suggests that they have significant potential to produce alginate oligosaccharides with a lower degree of polymerization. Studied alginate lyases having similar substrate specificity were produced by marine *Microbulbifer* sp. ALW1 [27], marine *Vibrio* sp. NJ-04 [43], and *Marinimicrobium* sp. H1 [30]. The results of the substrate specificity of alginate lyase were confirmed by the use of HPLC to determine the change in molecular weight of oligosacharides produced by the hydrolysis of three substrates, including sodium alginate, PolyM, and PolyG, at 37 °C for 24 h.

For the control samples, 0.5% of sodium alginate, PolyM, and PolyG in 0.02 M Tris-HCl buffer were used. The degradation ability of AlyDS44 is shown in Table 2. It was shown that the control sample of sodium alginate had two main peaks: peak 1 and peak 2 had molecular weights of 577.8 KDa and 305 KDa, constituting 44.3% and 43.5%, respectively. Interestingly, in the first hour, the enzyme started to degrade sodium alginate to oligosaccharides with lower molecular weights of 350.6 KDa, 64.9 KDa, and 15.5 KDa, which accounted for 10.2%, 60.7%, and 21.8%, respectively. This demonstrates that it is possible for the degradation reaction to occur at room temperature and in a very short time (during the set up time for the HPLC).

After 24 h incubation at 37 °C, degradation products that had smaller sizes, of 18.7 KDa (15.2%), 6.6 KDa (15.3%), and 3.4 KDa (35%), were detected. These results indicate that AlyDS44 worked effectively on the sodium alginate substrate and degraded it into oligosaccharides of approximately 30-times-lower molecular weight. In comparison to sodium alginate, PolyM and PolyG had lower initial molecular weights of 17.7 KDa (79%) and 14 KDa (80%), respectively. After the commencement of incubation (1 h), their chains were cut into smaller sizes of 15.8 KDa (65.5%) for PolyM and 10.9 KDa (65.8%) for PolyG. They were subsequently broken down to MW sizes of 9.8 KDa (54.7%) for PolyM and 3.1 KDa (51%) for PolyG after 24 h incubation. Moreover, the smallest molecular weight of degradation products detected from the reaction between AlyDS44 and PolyG substrate was 0.5 KDa, which accounted for 34.5% of the total. Therefore, it was concluded that AlyDS44 possesses strong activity against alginate substrate and could degrade PolyG more effectively than PolyM.

## 3. Materials and Methods

### 3.1. Materials

The seaweed used in this study was *Durvillaea potatorum* (bull kelp), which belongs to the brown seaweed group. Bull kelp was collected at Rivoli Bay (latitude: 37°31′00″ S, longitude: 140°04′12″ E), Beachport, South Australia and was used as a carbon source for the growth of actinobacteria and as a substrate for producing alginate lyase.

### 3.2. Cultivation and Identification of Actinobacterium DS44

Two loopfuls of spore mass of isolated *Streptomyces* strain DS44 grown on HPDA plates for at least 7 days were inoculated into 50 mL of IM22 inoculum medium (15 g/L glucose, 2 g/L calcium carbonate, 5 g/L sodium chloride, 15 g/L soyatone, 5 g/L Pharmamedia) in 250 mL Erlenmeyer flasks and then cultured at 27 °C, with shaking at 150 rpm. After three days, the inoculum was transferred to 50 mL production medium in 250 mL Erlenmeyer flasks at 5% (*v*/*v*). The production medium was prepared by adding 2% bull kelp powder (prepared by grinding and sieving) to 5 g/L peptone, 1 g/L yeast extract, 1 g/L glucose, 20 g/L NaCl, 2 g/L K_2_HPO_4_, 0.2 g/L MgSO_4_.7 H_2_O, and 5 g/L NH_4_Cl. The medium was sterilized by autoclaving at 121 °C for 15 min. After inoculation, the flasks were incubated at 27 °C on a rotary shaker at 150 rpm for 7 days.

Total genomic DNA of strain DS44 was isolated using a modified CTAB (hexadecyl trimethyl ammonium bromide) protocol based on the method described by Doyle and Doyle [47]. DNA quantification was checked using a NanoDrop spectrophotometer (ND-8000 spectrophotometer). The extracted DNA was sent to The United States Department of Agriculture (Peoria, IL, USA) to sequence the whole genome.

### 3.3. Purification of Alginate Lyase

#### 3.3.1. Concentration of Crude Enzyme

The broth cultures were centrifuged at 10,000× *g* for 20 min at 4 °C. The protein was then precipitated from the supernatant by stepwise ammonium sulfate addition, and the crude enzyme was collected at 60% saturation overnight at 4 °C. The solution was centrifuged at 10,000× *g* at 4 °C for 15 min. The precipitate was collected and dissolved in 20 mM Tris-HCl buffer. It was dialyzed by using dialysis tubing (cellulose membrane; 14 kDa molecular weight cut off, Sigma Aldrich, Sydney, Australia) in 1 L of the same buffer at 4 °C for desalting. The buffer was changed every two hours for 6 h. After dialysis, the supernatant was used as a partially purified enzyme for testing enzyme activity.

#### 3.3.2. Ion Exchange Chromatography and Size Exclusion Chromatography

The concentrated enzyme was applied to an anion exchange column (Mono-Q^TM^ column, 5/50 GL, GE Healthcare/Cytiva, Marlborough, MA, USA) attached to a FPLC (ÄKTA™ pure, Cytiva, Marlborough, MA, USA). The column had been previously equilibrated with buffer A (20 mM Tris-HCl pH 8.0). The pump was run at a flow rate of 1 mL/min. The enzyme was eluted with a linear gradient of 0–1 M NaCl in buffer B (1 M NaCl, 20 mM Tris-HCl pH 8.0). The fractions obtained from anion exchange chromatography were evaluated for enzyme activity and protein concentration using a Pierce BCA Protein Assay Kit (Thermo Scientific, Waltham, MA, USA). The fractions possessing the highest specific enzyme activity were pooled and kept in the freezer at −20 °C before loading onto a Superdex^TM^ 75 column (10/300 GL, GE Healthcare/Cytiva, Marlborough, MA, USA). The concentrated pooled fraction (1 mL) was loaded onto the column and elution was performed with 20 mM Tris HCl (pH 8.0) buffer containing 0.1 M NaCl at a flow rate of 0.5 mL/min. The fractions (1 mL) obtained from the size exclusion column were evaluated for their enzyme activity and protein concentration. The fractions with a high specific enzyme activity were stored at −20 °C and subsequently used for characterization studies.

### 3.4. Alginate Lyase Activity Assay

The activity of AlyDS44 was determined by a colorimetric method, based on the amount of reducing sugar released using 3-5-dinitrosalicylic acid (DNS) reagent [48]. A reaction mixture containing 0.5 mL partially purified enzyme and 0.5 mL substrate (0.5%, sodium alginate) in 0.02 M Tris-HCl buffer (pH 7.5) was incubated at 37 °C for 2 h. The reaction was stopped by adding 1 mL of 3-5-dinitrosalicylic acid reagent, and then the mixture was boiled for 5 min. The absorbance resulting from the reaction with the released sugars was measured at 540 nm in a UV Spectrophotometer. The concentration of reducing sugars released into the medium was determined by using D-glucose as standard. One unit of enzyme activity was defined as the amount of enzyme required to produce 1 µmol of reducing sugar per min.

The relative activity was calculated as the percentage ratio of activity at a given pH/temperature to the activity at optimum pH/temperature.

Statistical analysis: all results are based on an average of three experiments. The results were analyzed statistically by one-way ANOVA in SPSS.

### 3.5. Characterisation of Alginate Lyase

#### 3.5.1. SDS-Polyacrylamide Gel Electrophoresis

The apparent molecular weight of AlyDS44 was determined by SDS-Polyacrylamide Gel Electrophoresis. A 4–20% Criterion^TM^ TGX Stain-Free^TM^ Precast Gel (10 or 18 wells, Bio-Rad) was used for running SDS-PAGE. Reference proteins (5 uL) of Precision Plus Protein Unstained Standards (Bio-Rad, Herules, CA, USA), which are a mixture of ten Strep-tagged, recombinant proteins (10–250 kD), including three reference bands (25, 50, and 75 kD), were used as molecular mass markers. Next, 5 µL of 4× loading buffer was added to 15µL of sample, then the mixture was boiled at 95 °C for two minutes, centrifuged briefly and loaded onto a 4–20% stain free gel (Bio-Rad, Hercules, CA, USA). The gel was run at 300 V for approximately 20 min using a criterion tank (Bio-Rad) containing 1× running buffer. After electrophoresis, the gel was placed on a Stain Free Tray and the gel was imaged using automated exposure by a Gel Doc EZ imager (Bio-Rad, Hercules, CA, USA) to visualize protein bands.

#### 3.5.2. Amino Acid Sequencing of AlyDS44

The AlyDS44 band was excised from the gel, washed, reduced/alkylated, and digested with Trypsin, as described previously [28]. Alginate lyase peptides were then sequenced on a 5600+ qTOF mass spectrometer (AB Sciex, Framingham, MA, USA) fitted with an Ekspert nano LC 415 high performance liquid chromatography (HPLC) (Eksigent, AB Sciex, Dublin, CA, USA) Peptides were loaded onto a C18 trap and eluted onto a Nikkyo Technos (Tokyo, Japan) 15-cm capillary spray column containing 5 micron C18 beads using an acetonitrile gradient [49]. The spectra were analyzed by Pilot Protein software (version 4.5) (www.sciex.com, accessed on 6 October 2021) using protein mode against the *Streptomyces* amino acid database (UniProt).

#### 3.5.3. Multiple Sequence Alignment

The matched peptide sequences were analyzed using Clustal Omega (EMBL-EBI) to identify sequences with homology to reported alginate lyases. Amino acid sequences of alginate lyase AlyVI from *Vibrio* sp. QY101 (Genbank: AAP45155.1), AlyVOA from *Vibrio* sp. O2 (Genbank: ABB36771.1), A9mT from *Vibrio* sp. A9m (Genbank: BAH79132.1), AlgNJU-03 from *Vibrio* sp. NJU-03 (ASA33933.1), and ALG5 from *Streptomyces* sp. ALG5 (Genbank: ABS59291.1) were used for multiple sequence alignment with AlyDS44.

### 3.6. ESI-MS Analysis of Alginate Lyase Degradation Products

Electrospray-ionization mass spectroscopy (ESI-MS; Waters Synapt HDMS) was used to analyze the mass/charge ratio of the alginate lyase degradation products. Samples were run in negative ionization mode. A reaction containing 0.5% sodium alginate in 0.02 M Tris-HCl buffer (pH 8.0) was incubated with purified enzyme at 37 °C for 24 h. Samples were diluted in 1:50 methanol and then injected into electrospray. Mass calibration was performed via injection of sodium chloride. Mass range of the scans was from m/z 150 to 2000.

### 3.7. Effect of Various Factors and Compounds on Alginate Lyase Activity

#### 3.7.1. Determination of the Optimal pH

The reaction solution containing purified enzyme (100 µL) and substrate (500 µL, 0.5% sodium alginate in 0.02 M Tris-HCl buffer) with a range of pH values (4.5, 5.5, 6.5, 7.5, 8.5, 9.5, and 10.5) was incubated for 2 h at 37 °C. The enzyme activity was measured by the DNS assay.

#### 3.7.2. Determination of the Optimal Temperature

The purified enzyme was incubated with the substrate (0.5% sodium alginate) in 0.02 M Tris-HCl buffer at 15 °C, 25 °C, 35 °C, 45 °C, 65 °C, and 75 °C for 2 h.

#### 3.7.3. Determination of Salt Tolerance

The purified enzymes were incubated with the substrate (0.5% sodium alginate) in 0.02 M Tris-HCl buffer at NaCl concentrations of 0.2 M, 0.4 M, 0.6 M, 0.8 M, and 1.0 M for 2 h at 37 °C.

#### 3.7.4. Influence of Multivalent Metal Ions on Enzyme Activity

The reaction mixture containing purified enzyme and substrate (0.5% sodium alginate) in 0.02 M Tris-HCl buffer and 0.2 mL of 5 mM the appropriate salt (CaCO_3_, CoCl_2_, CuSO_4_, FeSO_4_, FeCl_3_, MgSO_4_, MnCl_2_, ZnCl_2_) was incubated for 2 h at 37 °C.

### 3.8. Substrate Specificity

To assay the hydrolytic behavior of purified alginate lyase, three different substrates, including sodium alginate, polymannuronic (PolyM), and polyguluronic (PolyG), were used. Each reaction contained 100 µL each of purified enzyme and 500 µL 0.5% of each substrate in 0.02 M Tris-HCl buffer (pH 8.0). They were incubated at 37 °C for 2 h and then enzyme activity was evaluated by DNS assay.

### 3.9. HPLC Analysis of Alginate Hydrolysis

To identify alginate hydrolysis ability of AlyDS44, the molecular weight of hydrolysis products was determined by HPLC (Shimadzu, Kyoto, Japan) with a GPC/SEC column: PL aquagel-OH Mixed-H 8 um 300 × 7.5 mm (6 kDa–10 MDa) and PL aquagel-OH 30 8 µm 300 × 7.5 mm (0.1–60 kDa). The mobile phase was 0.1 M sodium nitrate (isocratic). The standards and tested samples were dissolved in mobile phase. For preparing the sample, 100 µL of purified enzyme was incubated with 500 µL of substrate, including 0.5% sodium alginate, 0.5% PolyM, and 0.5% PolyG in 0.02 M Tris-HCl buffer (pH 8.0) at 37 °C. Then, 100 µL of sample was taken at 1 h and 24 h incubation and diluted with 100 µL of 0.1 M sodium nitrate. The sample was centrifuged at 19,000× *g* for 5 min and then 100 µL of sample was placed into an HPLC vial. A 50 µL sample was injected and run through the column with a flow rate of 1.0 mL/min. The peaks were detected using a refractive index (RI) detector.

## 4. Conclusions

In this study, the purification and characterization of a novel alginate lyase AlyDS44 was accomplished. The successful purification of the alginate lyases was achieved by combining ammonium sulfate precipitation, anion exchange chromatography, and size exclusion chromatography. AlyDS44 has a molecular weight of 28.6 KDa. The alginate lyase gene was composed of 780 bp, encoding 260 amino acid residues. AlyDS44 could degrade both PolyM and PolyG, indicating that it is bifunctional alginate lyase; however, it preferentially degraded PolyG. This characteristic suggests that AlyDS44 has the potential to produce alginate oligosaccharides with lower degrees of polymerization. Moreover, based on the highly conserved regions of their amino acid sequences, this enzyme was classified to the PL7 family. AlyDS44 could hydrolyze sodium alginate to produce alginate disaccharides and trisaccharides as the main end products, indicating that it was an endo-type alginate lyase. Furthermore, this enzyme showed the highest activity at pH 8.5, when the degradation reaction was performed at 45 °C. AlyDS44 also exhibited an unchanged activity at high concentrations of NaCl, which led it to be grouped into salt-tolerant alginate lyases, isolated from marine bacteria. AlyDS44 was more active in the presence of Mn^2+^, Co^2+^, and Fe^2+^ and inhibited by Zn^2+^ and Cu^2+^. Finally, the amino acid sequence of DS44 showed more than 10% difference in amino acid residues from a closely related alginate lyase from *Streptomyces* sp. ALG-5, which demonstrates that AlyDS44 from *Streptomyces luridiscabiei* is a novel enzyme. To sum up, AlyDS44 is a candidate to be an industrial enzyme for the efficient production of alginate oligosaccharides.

## Figures and Tables

**Figure 1 marinedrugs-19-00590-f001:**
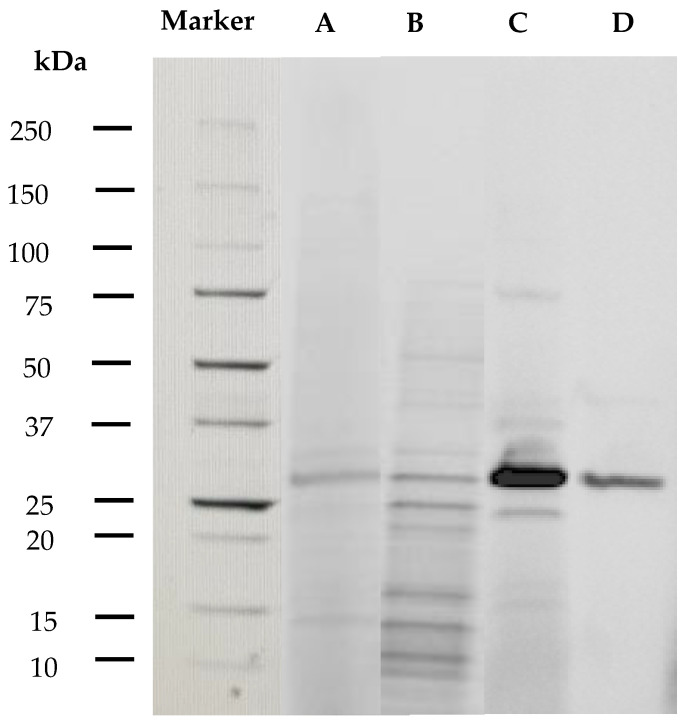
SDS–PAGE of various fractions obtained during purification: marker, crude enzyme (**A**), dialyzed enzyme after ammonium sulfate precipitation (**B**), active fractions of purified enzyme by anion exchange chromatography (**C**), and size exclusion chromatography (**D**).

**Figure 2 marinedrugs-19-00590-f002:**
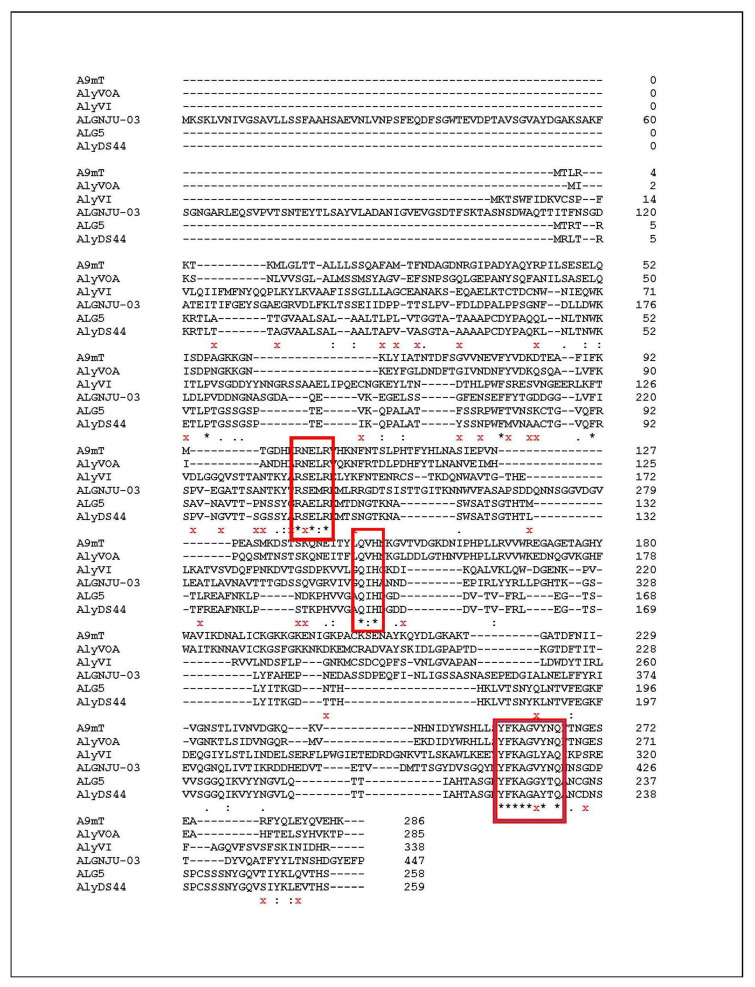
Multiple sequence alignment of amino acid sequences of AlyDS44 alginate lyases with other alginate lyases. Five alginate lyases: AlyVI from *Vibrio* sp. QY101 (AAP45155.1), AlyVOA from *Vibrio* sp. O2 (ABB36771.1), A9mT from *Vibrio* sp. A9m (BAH79132.1), AlgNJU-03 of *Vibrio* sp. NJU-03 (ASA33933.1), and ALG-5 from *Streptomyces* sp. ALG5 (ABS59291.1) were used for comparison. Boxes with * below the sequences are presumed to form an active center. x -denotes different amino acid residue compared to ALG-5.

**Figure 3 marinedrugs-19-00590-f003:**
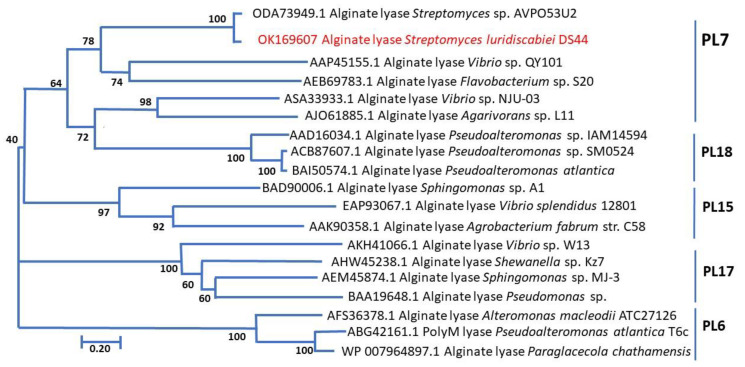
Phylogenetic tree using the neighbor-joining method of AlyDS44 alginate lyase with other reported alginate lyases. The relationship between AlyDS44 (in red) and different PL families is indicated. The numbers at the branching points are the percentages of occurrence in 1000 bootstrapped trees. The bar indicates a distance of 0.2 substitutions per site.

**Figure 4 marinedrugs-19-00590-f004:**
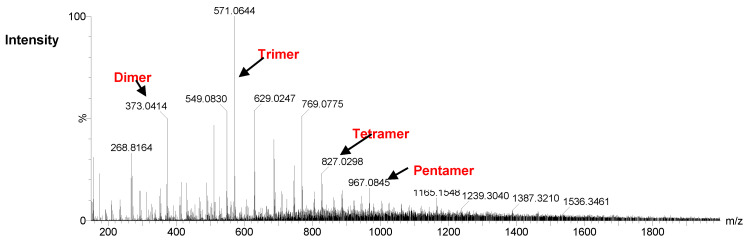
ESI-MS analysis of the hydrolysis products by AlyDS44 alginate lyases.

**Figure 5 marinedrugs-19-00590-f005:**
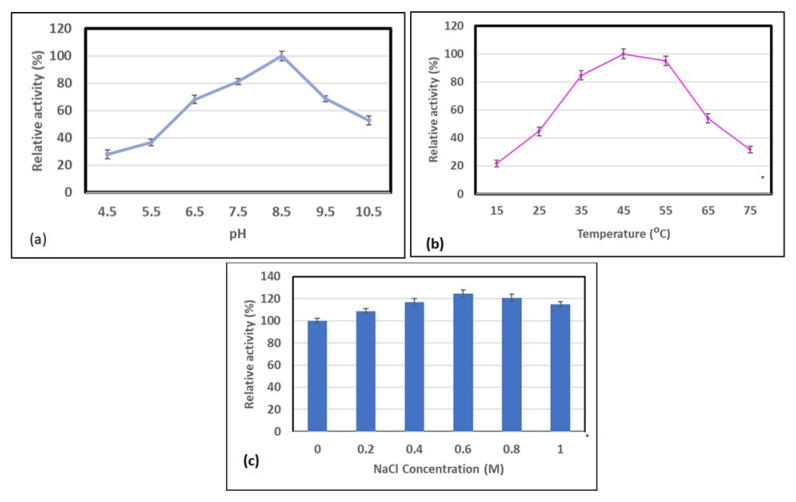
Optimal pH (**a**), temperature (**b**), and NaCl concentration (**c**) of AlyDS44. The highest activity was set as 100% for pH and temperature. Each value represents the mean of triplicates ± standard deviation.

**Figure 6 marinedrugs-19-00590-f006:**
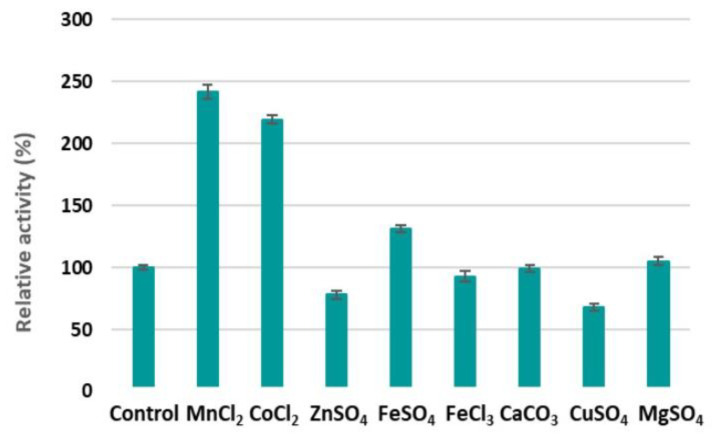
The effect of metal ions (5 mM) on the activity of AlyDS44. The activity of the control (no metal ion) was taken as 100%. Each value represents the mean of triplicates ± standard deviation.

**Figure 7 marinedrugs-19-00590-f007:**
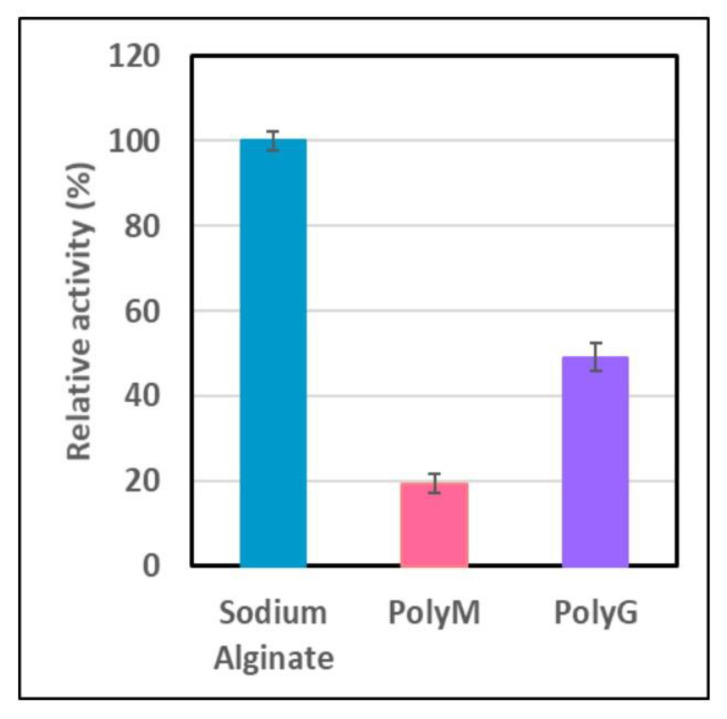
Hydrolysis of three different substrates by AlyDS44.

**Table 1 marinedrugs-19-00590-t001:** Summary of purification of AlyDS44 alginate lyase (n = 3).

Purification Steps	Volume (mL)	Total Protein (mg)	Total Activity (U)	Specific Activity (U/mg)	Yield (%)	Purification (Fold)
Culture broth	300	604.5	695.9	1.15	100	1
60% Ammonium sulfate precipitate (after dialysis)	12	13.52	370.8	27.54	53.3	23.9
Mono-Q Active fractions	1	0.327	28.88	93.78	4.15	81.5
Superdex 75 Purified product	1	0.115	12.49	108.6	1.79	94.4

**Table 2 marinedrugs-19-00590-t002:** The change of molecular weight of three substrates after incubation with AlyDS44 for 1 h and 24 h. Only the three highest MW peaks are listed.

	Peak 1		Peak 2		Peak 3	
	%	MW (kDa)	%	MW (kDa)	%	MW (kDa)
**Sodium alginate**						
**Control**	44.3	577.8	43.5	305.0	0.2	2.1
**1 h**	10.2	350.6	60.7	64.9	21.8	15.5
**24 h**	15.2	18.7	15.3	6.6	35.0	3.4
**Poly M**						
**Control**	79.9	17.7	2.9	2.2	1.7	1.0
**1 h**	65.5	15.8	3.7	2.2	1.6	1.0
**24 h**	54.7	9.8	8.3	4.1	5.8	2.2
**Poly G**						
**Control**	80.9	14.4	0.4	2.1	16.7	0.6
**1 h**	65.8	10.9	2.5	2.1	31.9	0.5
**24 h**	51.0	3.1	14.3	2.1	34.5	0.5

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
