# Peer review of "Purification and Characterization of a Novel Alginate Lyase from a Marine Streptomyces Species Isolated from Seaweed"

_marinedrugs, 2021, doi:10.3390/md19110590_

Round 1

Reviewer 1 Report

The authors purified and characterized a alginate lyase from marine actinobacterium, Streptomyces luridiscabiei. It is a meaningful work and could be accepted for publication in Marine Drugs. However, there are some major concerns needed to be addressed.

  1. The sequence similarity of AlyDS44 with other enzymes should be described and discussed.
  2. Line 98, Bull Kelp?Kelp powder or something else?
  3. Line 128, It is impossible to calculate the MW of AlyDS44 only by SDS-PAGE, therefore it should be expressed that the MW of AlyDS44 lied  between 25~37 KDa.
  4. Line 160, the alginate lyase could be classified into 14 families, namely PL5, PL6, PL7, PL8, PL14,PL15, PL17, PL18, PL31, PL32, PL34, PL36, PL39 families.
  5. Line 162, contains?
  6. Line 215~216, "AlyDS44 alginate lyase presented over 50% relative activity in the temperature range  of 35oC to 55oC, and then dropped dramatically to approximately 35% at 65oC" It is not clear and please rephrase this sentence.
  7. Line 240, seaweed alginate?
  8. The whole figures in manuscript should be replaced with more clear ones.
  9. The language and grammar should be extensively edited by a professional and English-native expert.

Author Response

Responses to Reviewer 1:

WE thank Reviewer 1 for their comments which we have addressed completely as follows:

  1. The sequence similarity of AlyDS44 with other enzymes should be described and discussed.
    This has been addressed and we have indicated in Fig.2 how the AlyDS44enzyme differs in amino acid sequence from the closest enzyme.
  2. Line 98, Bull Kelp?Kelp powder or something else?

This has been changed to Bull Kelp powder.

3. Line 128, It is impossible to calculate the MW of AlyDS44 only by SDS-PAGE, therefore it should be expressed that the MW of AlyDS44 lied  between 25~37 KDa
This has been changed to25~29KDa..

4. Line 160, the alginate lyase could be classified into 14 families, namely PL5, PL6, PL7, PL8, PL14,PL15, PL17, PL18, PL31, PL32, PL34, PL36, PL39 families.
This has been changed to include all 14 families

5. Line 162, contains?
This has been corrected.

6. Line 215~216, "AlyDS44 alginate lyase presented over 50% relative activity in the temperature range  of 35oC to 55oC, and then dropped dramatically to approximately 35% at 65oC" It is not clear and please rephrase this sentence.
This has been corrected and the duplicate paragraph removed

7. Line 240, seaweed alginate?
This has been changed to remove seaweed and includes PolyM and PolyG.

8. The whole figures in manuscript should be replaced with more clear ones.

The figures which were not clear have been replaced (Fig .2) with a more clear Figure or removed (Fig. 8).

9. The language and grammar should be extensively edited by a professional and English-native expert.
Extensive editing has been carried out as requested.

Reviewer 2 Report

The authors describe the discovery and characterization of a novel alginate lyase, which is an important enzyme for the industrial degradation of seaweed alginate. The manuscript is well written, comprehensive, and clear. The experiments performed are sufficiently described and the results afforded a new enzyme suitable for large applications.

There are only minor issues concerning this manuscript:

  1. Paragraphs (l. 209-221) concerning the optimal temperatures are a little bit redundant, please revise the text.
  2. Please check the font inconsistencies in paragraph 3.3.2
  3. It would be nice to show the structure of alginate on a figure in the introduction.
  4. Figure 8 - the peaks should be described, with respect to their MW. Where are the low-MW products on the chromatogram?

Author Response

Responses to Reviewer 2:

WE thank Reviewer 2 for their comments which we have addressed completely as follows:

1. Paragraphs (l. 209-221) concerning the optimal temperatures are a little bit redundant, please revise the text.

 The text has been revised and the duplicate paragraph removed.

2. Please check the font inconsistencies in paragraph 3.3.2
This has been corrected.

3. It would be nice to show the structure of alginate on a figure in the introduction.
We agree with the reviewer but as we do not have a chemist it is proven a little difficult to obtain the software.

4. Figure 8 - the peaks should be described, with respect to their MW. Where are the low-MW products on the chromatogram?

WE decided to remove Fig 8 as the MWs have been described in Table 2, and the figure had poor resolution. The lowest MW detected on the HPLC Chromatogram is 0.5 KDa which is seen for the PolyG hydrolysate.

Reviewer 3 Report

A novel alginate lyase from Streptomyces luridiscabiei was isolated and characterized. This bacterium had not been investigated before for AlgPL. It is concluded that the enzyme belongs to bifunctional PL family 7, showing preferential cleavage of poly-G. The products were characterized by MS. The work is well done.

I suggest minor revision of the introduction: The references are not up-to-date, authors may consider to add citation of recent reviews, see e.g. Cheng et al. (2020), http://dx.doi.org/10.1016/j.ijbiomac.2020.07.199; Li et al. (2021), http://dx.doi.org/10.1080/07388551.2021.1898330

Otherwise, the paper is well written and should be published in Marine Drugs.

Author Response

We thank Reviewer 3 and have included the suggestion for 2 additional references

I suggest minor revision of the introduction: The references are not up-to-date, authors may consider to add citation of recent reviews, see e.g. Cheng et al. (2020), http://dx.doi.org/10.1016/j.ijbiomac.2020.07.199; Li et al. (2021), http://dx.doi.org/10.1080/07388551.2021.1898330

These have been added as ref 25 and 26

Round 2

Reviewer 1 Report

The manuscript has been well revised and could be accepted for publication.